# The Roles of Fibrinolytic Factors in Bone Destruction Caused by Inflammation

**DOI:** 10.3390/cells13060516

**Published:** 2024-03-15

**Authors:** Yosuke Kanno

**Affiliations:** Department of Molecular Pathology, Faculty of Pharmaceutical Science, Doshisha Women’s College of Liberal Arts, 97-1 Kodo Kyotanabe, Kyoto 610-0395, Japan; ykanno@dwc.doshisha.ac.jp; Tel.: +81-0774-65-8629

**Keywords:** fibrinolytic factors, inflammation, bone destruction, osteoblasts, osteoclasts, osteocytes

## Abstract

Chronic inflammatory diseases, such as rheumatoid arthritis, spondyloarthritis, systemic lupus erythematosus, Crohn’s disease, periodontitis, and carcinoma metastasis frequently result in bone destruction. Pro-inflammatory cytokines such as tumor necrosis factor-α (TNF-α), interleukin-1β (IL-1β), IL-6, and IL-17 are known to influence bone loss by promoting the differentiation and activation of osteoclasts. Fibrinolytic factors, such as plasminogen (Plg), plasmin, urokinase-type plasminogen activator (uPA), its receptor (uPAR), tissue-type plasminogen activator (tPA), α2-antiplasmin (α2AP), and plasminogen activator inhibitor-1 (PAI-1) are expressed in osteoclasts and osteoblasts and are considered essential in maintaining bone homeostasis by regulating the functions of both osteoclasts and osteoblasts. Additionally, fibrinolytic factors are associated with the regulation of inflammation and the immune system. This review explores the roles of fibrinolytic factors in bone destruction caused by inflammation.

## 1. Introduction

Bone homeostasis is regulated by the balance between osteoblast/osteocyte-mediated bone formation and osteoclast-mediated bone resorption. Osteoclast differentiation and maturation are induced by macrophage colony-stimulating factor (M-CSF) and receptor activator of NF-κB ligand (RANKL) [1]. Chronic inflammatory diseases, such as rheumatoid arthritis, spondyloarthritis, systemic lupus erythematosus, Crohn’s disease, periodontitis, and carcinoma metastasis frequently cause bone destruction [2]. Inflammatory mediators, such as tumor necrosis factor (TNF)-α and interleukin 1β (IL-1β), IL-6, IL-17, parathyroid hormone (PTH), and prostaglandin E_2_ (PGE_2_) affect osteoblasts and osteoclasts, resulting in bone destruction through osteoclast differentiation and activation [3,4,5]. Additionally, inflammatory mediators cause osteocyte apoptosis and result in decreased bone mass.

In the fibrinolytic system, plasminogen (Plg) undergoes conversion into plasmin through the action of urokinase-type plasminogen activator (uPA) and its receptor (uPAR), or tissue-type plasminogen activator (tPA), leading to the dissolution of fibrin. In contrast, α2-antiplasmin (α2AP) serves as the primary inhibitor of plasmin, while plasminogen activator inhibitor-1 (PAI-1) can bind to both tPA and uPA, thereby inhibiting the generation of plasmin [6]. In recent years, numerous studies have demonstrated that these fibrinolytic factors not only degrade fibrin but also serve various functions, including growth factor activation, cytokine production, cell differentiation, cell proliferation, and cell migration. They play pivotal roles in biological processes such as inflammation, tissue remodeling, angiogenesis, and the immune system [6,7,8,9]. Fibrinolytic factors are expressed in bone cells including osteoclasts, osteoblasts, and osteocytes and are considered essential in maintaining bone homeostasis by regulating the functions of osteoclasts, osteoblasts, and osteocytes [10]. This review describes the role of fibrinolytic factors in maintaining bone homeostasis and contributing to inflammatory bone destruction.

## 2. Bone Remodeling and Inflammatory Bone Destruction

Bone remodeling is regulated through the coordinated actions of osteoclasts, osteoblasts, and osteocytes [4]. Osteoclasts are multinucleated cells that arise from mononuclear precursors of monocytes and macrophages in response to RANKL and M-CSF. They are responsible for bone resorption [11,12,13]. Additionally, pro-inflammatory cytokines, such as TNF-α, IL-1β, IL-6, and IL-17, promote osteoclast function through multiple signaling pathways, including NF-κB and mitogen-activated protein kinases (MAPKs: extracellular signal-regulated kinase (ERK), p38, c-Jun terminal kinase (JNK)) [3,14,15,16]. PGE_2_ enhances RANKL-induced osteoclast differentiation [17]. Moreover, matrix metalloproteinase-9 (MMP-9) and MMP-13 are involved in the process of bone resorption through differentiation and activation of osteoclasts [18,19,20]. In contrast, it has been reported that interferon-γ (IFN-γ), IL-4, IL-5, IL-10, and IL-13 inhibit osteoclastogenesis [21,22,23,24]. On the other hand, osteoprotegerin (OPG) acts as a decoy receptor and inhibits RANKL/RANK signaling [4]. Osteoblasts are derived from mesenchymal stem cells (MSCs) and play an important role in bone formation. Osteoblasts produce extracellular proteins, such as type I collagen and osteocalcin, and mediate bone mineralization. Conversely, osteoblasts regulate osteoclast differentiation through the production of several osteoclastogenic factors, including RANKL, M-CSF, TNF-α, IL-1β, and PGE_2_, as well as anti-osteoclastogenic factors, including OPG [2,25]. In contrast, osteoclasts produce several osteogenesis factors, including collagen triple helix repeat containing 1 (CTHRC1), sphingosine-1-phosphate (S1P), and C3, as well as anti-osteogenesis factors, including semaphorin-4D (SEMA4D) [26,27,28,29,30]. Additionally, osteoclast-mediated bone resorption releases osteogenesis factors, such as transforming growth factor-β (TGF-β) and insulin-like growth factor 1(IGF-1) [26]. Thus, bone homeostasis is maintained through osteoclast–osteoblast communication (Figure 1). Osteocytes are cells that originate from osteoblasts and become embedded in the mineralized bone matrix [4]. They play a crucial role in both bone formation and the maintenance of the matrix. Osteocyte apoptosis has been associated with osteoclastic bone resorption. Osteocyte apoptosis is linked to several pathological conditions, such as aging, inflammation, unloading/disuse, fatigue/microdamage, excess glucocorticoids (GCs), and estrogen (Es), or androgen (As) deficiency [31]. Pro-inflammatory mediators, such as high mobility group box 1 (HMGB1), TNF-α, IL-6, and advanced glycation end products (AGEs), induce osteocyte apoptosis. Apoptotic osteocytes, in turn, produce pro-inflammatory mediators such as TNF-α, IL-1β, IL-6, HMGB1, and AGEs [31]. These factors derived from apoptotic osteocytes may contribute to osteoclastogenesis, and osteocyte apoptosis is implicated in inflammatory bone destruction (Figure 1).

## 3. The Role of Fibrinolytic Factors in Bone Homeostasis and Inflammatory Bone Destruction

It is known that inflammation causes activation of the coagulation system, resulting in the formation of fibrin clots. In contrast, fibrin deposition acts as a stimulator of inflammatory responses and is associated with chronic inflammatory diseases [32]. Inflammation and coagulation are two closely related processes. Fibrin deposition is part of normal acute inflammation [33], and fibrin mediates the inflammatory response through a variety of cellular receptors, including Toll-like receptor (TLR) [34,35]. The role of fibrin as a driver of inflammatory bone loss in osteoporosis is supported by evidence showing that all bone pathologies (and corresponding inflammatory markers) are reduced in Plg-deficient mice by crossing these animals with fibrinogen-deficient mice or expressing a mutant form of fibrinogen that retains clotting function but lacks the αMβ2-binding motif [36]. Additionally, fibrin promotes osteoclastogenesis through integrin αMβ2 and induces RANKL expression in osteoblasts. In contrast, platelet-rich fibrin accelerates wound healing and bone regeneration [37]. Thus, fibrin plays an important role in bone homeostasis. The fibrinolytic system, responsible for dissolving fibrin, is associated with vascular homeostasis, tissue remodeling, and the immune and inflammatory response. Fibrinolytic factors, including Plg, uPA, uPAR, tPA, α2AP, and PAI-1, have various functions other than fibrin degradation, and they are expressed in both osteoclasts and osteoblasts. These factors play a regulatory role in the functions of both osteoclasts and osteoblasts (Figure 2). On the other hand, the levels of fibrinolysis markers such as D-dimer and fibrin degradation products (FDP), as well as fibrinolytic factors in samples from patients with chronic inflammatory diseases like rheumatoid arthritis [38,39,40,41], systemic lupus erythematosus [42,43,44,45,46,47,48,49], Crohn’s disease [50,51,52,53], and periodontitis [54,55,56], are higher than those in healthy controls (Figure 3). These factors are associated with the pathology of chronic inflammatory diseases and may play a critical role in the context of inflammatory bone destruction.

Fibrinolytic factors, including Plg, uPA, uPAR, tPA, α2AP, and PAI-1, have been detected in samples from patients with chronic inflammatory diseases, including rheumatoid arthritis, systemic lupus erythematosus, Crohn’s disease, and periodontitis.

### 3.1. Plasminogen (Plg) and Plasmin

Plg acts as the precursor or zymogen of plasmin and consists of an N-terminal heavy chain and a C-terminal light chain, with the latter containing the proteolytic active site [7]. Plg can be found in interstitial tissues due to plasma exudation during inflammation and tissue injury. It can bind to fibrin as well as various receptors, including the plasminogen receptor (Plg-R_KT_), enolase-1, the heterotetrametric complex Annexin A2-S100A10, and histone H2B [57,58]. Plasminogen is converted to plasmin by uPA/uPAR or tPA. Plasmin plays a crucial role in regulating fibrinolysis, activating growth factors (such as TGF-β, vascular endothelial growth factor (VEGF), insulin-like growth factor-binding protein 5 (IGFBP-5), basic fibroblast growth factor (bFGF), and pro-brain-derived neurotrophic factor (pro-BDNF)), MMPs (MMP-1, MMP-3, MMP-9, and MMP-13), extracellular matrix (ECM) degradation, activating protease-activated receptors (PARs) (PAR-1 and PAR-4), and hormone processing. Additionally, plasmin mediates various cellular functions, cytokine production, apoptosis, tissue remodeling, and the inflammatory response through multiple mechanisms [59,60,61,62]. 

Plg deficiency has been shown to decrease trabecular and cortical bone mineral densities, and the administration of plasmin rescues these effects in mice [63]. Plg deficiency also delays bone repair, leading to decreased bone formation [64]. Additionally, a reduction in the number of osteoblasts and macrophages occurs after the bone becomes defective in mice [65]. Furthermore, Cole et al. have demonstrated that Plg deficiency leads to fibrin deposition within the bone, and this fibrin is associated with inflammation and bone destruction in mice [36]. On the other hand, plasmin has been found to attenuate lipopolysaccharide (LPS)-induced osteoclastogenesis through the PAR-1/AMP-activated protein kinase (AMPK) pathway [66], and it induces OPG production through the ERK1/2 and p38 pathways in osteoblasts [63]. 

Plg and plasmin-α2AP (PAP) have been detected in samples from patients with chronic inflammatory diseases, including rheumatoid arthritis, systemic lupus erythematosus, Crohn’s disease, and periodontitis (Figure 3). Furthermore, Plg and plasmin play crucial roles in regulating various steps in inflammation resolution, including macrophage reprogramming, neutrophil apoptosis, and efferocytosis [7,67]. TGF-β, bFGF, and VEGF regulate osteoblast and osteoclast differentiation [68,69]. Furthermore, plasmin activates MMP-9 and MMP-13, and the release and activation of these factors by plasmin may impact bone remodeling. Additionally, Plg and plasmin are associated with the production and release of pro-inflammatory cytokines such as TNF-α, IL-1β, and IL-6 as well as anti-inflammatory cytokines, such as TGF-β and IL-10 [70,71]. In contrast, it has been reported that plasmin causes inflammatory response, including cytokine production and chemotaxis [72,73]. The alteration in plasmin activity may impact the progression of chronic inflammatory diseases. Additionally, tranexamic acid, a lysine analogue that competes for lysine-binding sites in Plg, has been shown to exhibit anti-inflammatory effects by inhibiting plasmin in patients undergoing cardiac surgery [74]. TXA also suppresses the expression of inflammatory cytokines and inhibits inflammatory osteoclastogenesis. [75]. Moreover, plasmin induces mononuclear cell recruitment through PAR-1 activation [76], and PAR-1 activation induces M-CSF and IL-6 production through the phosphoinositide 3-kinase (PI3K)–Akt and MEK-ERK1/2 pathways in osteoblasts [77]. Plasmin also releases VEGF from the ECM through proteolysis [78]. VEGF, known as a regulator of survival, proliferation, and differentiation in bone cells, including osteoclasts, osteoblasts, and osteocytes, is decreased in apoptotic osteocytes [79]. Additionally, plasmin affects apoptosis in several types of cells [67,80]. The VEGF released by plasmin may impact osteocyte apoptosis. 

On the other hand, a deficiency in Plg-R_KT_ reduces the capacity of macrophages to phagocytose apoptotic neutrophils and impairs monocyte recruitment and migration during inflammation [7]. Additionally, the annexin A2/S100A10 complex positively regulates the expression of several cell surface receptors and ion channels [81], while annexin A2 and S100A10 modulate macrophage activation through TLR signaling [82,83]. Furthermore, the blockade of enolase-1 by a neutralizing antibody inhibits inflammation-enhanced osteoclast activity, mediates bone homeostasis, and impedes inflammation-induced migration and chemotaxis through a plasmin-related mechanism in mice [84]. Additionally, angiostatin is an internal fragment of Plg resulting from the proteolytic cleavage of Plg, and the generation of angiostatin is associated with uPA and tPA. Angiostatin inhibits cancer-induced bone destruction through a direct inhibition of osteoclast activity and generation in mice [85].

These data suggest that Plg, plasmin, and their receptors may play a pivotal role in bone homeostasis and inflammatory bone destruction through multiple mechanisms. 

### 3.2. Urokinase-Type Plasminogen Activator (uPA) and Its Receptor (uPAR)

uPA is a serine protease responsible for converting Plg to plasmin. The N-terminal domain of uPA, known as the N-terminal fragment (ATF), has the ability to bind to its receptor, uPAR. In contrast, its C-terminal domain is involved in catalytic activity [86]. uPAR is a glycosylphosphatidylinositol (GPI)-anchored protein composed of three domains (D1, D2, and D3). uPAR has the capacity to interact with several proteins, including uPA, integrins, low-density lipoprotein receptor-related protein 1 (LRP-1), and vitronectin (Vn), within the membrane, thereby regulating various signaling pathways [59,87]. uPAR undergoes cleavage between the D1 and D2 domains, as well as the GPI-anchor domain, by various enzymes, including uPA, plasmin, MMP-3, MMP-12, MMP-19, and MMP-25. This proteolytic cleavage results in the removal of DI, leading to the formation of a shorter uPAR form (DIIDIII-uPAR). This truncated form loses its ability to bind both uPA and Vn and to associate with integrins. However, when the SRSRY sequence (corresponding to amino acids 88–92) at the N-terminus is exposed, the truncated uPAR remains capable of interacting with *N*-formyl peptide receptors (FPRs). Both the full-length and cleaved forms of uPAR can be released from the cell surface in soluble forms known as suPAR [86,88].

uPA and uPAR play pivotal roles in regulating diverse cellular processes, including cell growth, inflammatory reactions, immune responses, bone metabolism, tissue remodeling, angiogenesis, adipose tissue development, fibrosis, and glucose metabolism. Their involvement is closely associated with the pathogenesis of various diseases, such as rheumatoid arthritis, cancer, fibrosis, and diabetes, through various signaling pathways, including Janus kinase (JAK) signal transducer and activator of transcription protein (STAT) and PI3K/Akt, focal adhesion kinase (FAK) [86,89,90,91,92]. uPA and uPAR mediate both pro- and anti-inflammatory responses, regulating cytokine production, cell invasion, chemotaxis, and phagocytosis through plasmin synthesis [93,94]. In contrast, uPAR interacts with various receptors, including integrins and LRP, participating in the initiation of the innate immune response by inducing cell migration and adhesion. Additionally, uPAR modulates TLR-2, -4, and -7 signaling, influencing inflammation (cytokine production and mediation of the NF-κB pathway), and immune responses, including the activation of macrophages and neutrophils [95,96,97]. The increase in uPAR expression is caused by hypoxia, infection, and inflammation, and the induction of uPAR is associated with the activation of transcription factors, such as NF-κB and AP-1 [98,99]. Blood suPAR levels correlate with the levels of established inflammatory biomarkers in humans, and uPAR exerts pro-inflammatory functions [98]. Additionally, uPA and uPAR have been detected in samples from patients with chronic inflammatory diseases, including rheumatoid arthritis, systemic lupus erythematosus, Crohn’s disease, and periodontitis (Figure 3). Deficiency in uPA and tPA increases bone formation and bone mass by promoting the accumulation of bone matrix proteins in mice [100]. Additionally, a study involving uPA- and tPA-deficient mice demonstrates that PAs play a crucial role in osteoclast-mediated bone digestion through proper integrin-dependent attachment to bone [101]. In contrast, uPA deficiency promotes LPS-induced inflammatory osteoclastogenesis and bone destruction in mice; uPA treatment attenuates inflammatory osteoclastogenesis through the plasmin/PAR-1/AMPK axis [66]. Furthermore, uPA deficiency has been shown to impede the early stages of bone repair in mice. In a mouse model of bone injury, a reduced bone repair process is observed at earlier time points, accompanied by a decrease in the number of macrophages and their phagocytic activity at the site of bone injury [65]. uPA also inhibits the inflammatory response in rats with a pulmonary thromboembolism model [102]. The synthesis of plasmin by PA plays a crucial role in both bone homeostasis and the regulation of inflammatory responses, contributing to the mediation of inflammatory bone destruction. 

On the other hand, the deficiency of uPAR has been linked to increased bone mass in mice. Furthermore, uPAR-deficient mice-derived osteoblasts exhibit heightened matrix mineralization and an earlier onset of alkaline phosphatase (ALP) activity. Notably, components of AP-1, such as JunB and Fra-1, are upregulated in uPAR-deficient mice-derived osteoblasts, in conjunction with other osteoblastic markers. On the resorptive side, the number of osteoclasts derived from uPAR-deficient monocytes has been found to decrease, resulting in increased bone mass in mice [103]. Moreover, the expression and release of M-CSF from osteoblasts are hindered by uPAR deficiency. uPAR plays a crucial role in regulating the formation, differentiation, and functional properties of macrophage-derived osteoclasts through the M-CSF binding receptor c-Fms/PI3K/Akt/NF-κB pathway [104]. Furthermore, uPAR deficiency attenuates MMP-9 expression in mice [90]. Thus, uPAR influences bone homeostasis by modulating the functions of both osteoblasts and osteoclasts. Furthermore, deficiency in uPAR and the blockade of uPAR by administering an anti-uPAR neutralizing antibody significantly attenuate LPS-induced inflammatory osteoclast formation and bone loss in mice [105]. uPAR plays a positive regulatory role in inflammatory osteoclast formation and bone loss through the integrin β3/Akt pathway [105]. Moreover, a uPA-derived peptide (Å6) has been shown to attenuate inflammatory osteoclast formation and bone loss in mice. Å6 achieves this attenuation of inflammatory osteoclast formation by inactivating the NF-κB and Akt pathways [106]. It has been reported that Å6 inhibits the interaction of uPA with uPAR [107]. The interaction between uPA and uPAR may play a role in mediating inflammatory osteoclast formation and bone loss through multiple mechanisms, including plasmin-dependent and -independent cell signaling. On the other hand, uPAR can interact with several factors, including integrins, VEGFR2, and caveolin-1 [8]. The binding of connective tissue growth factor (CTGF) to integrin αvβ3 induces osteocyte apoptosis through the activation of ERK1/2 [108]. Additionally, caveolin-1-dependent VEGFR2 activation promotes osteocyte viability [109]. uPA and uPAR may affect osteocyte survival. uPA and uPAR mediate various bone cells, including osteoclasts, osteoblasts, and osteocytes, through multiple plasmin-dependent and -independent mechanisms, and regulate bone homeostasis and inflammatory bone destruction.

### 3.3. Tissue-Type Plasminogen Activator (tPA)

tPA is responsible for converting Plg into plasmin. tPA is a mosaic protein comprising five distinct modules: a finger domain, an epidermal growth factor (EGF)-like domain, two Kringle domains, and a serine protease proteolytic domain. These modules are associated with the binding and activation of various substrates and/or receptors, including Plg, platelet-derived growth factor (PDGF), and the N-methyl-d-aspartate receptor (NMDAR) [110,111,112]. Deficiency in tPA and uPA increases bone formation and bone mass in mice [100]; PAs play a crucial role in osteoclast-mediated bone digestion through integrin [101]. In contrast, tPA deficiency delays the bone repair process in the femurs of mice, and tPA plays a crucial role in bone repair by facilitating osteoblast proliferation [113]. Furthermore, tPA inhibits the response to LPS through LRP1 in macrophages. Enzymatically active and inactive tPA demonstrate similar immune modulatory activity, and the administration of enzymatically inactive tPA blocks the toxicity of LPS in mice. LRP1 regulates various cellular cytokine signaling and suppresses TNF receptor (TNFR)/NF-κB activity, resulting in decreased expression of TNF-α [114]. Additionally, tPA mediates anti-inflammatory cell signaling and cytokine production through NMDAR and regulates innate immunity in macrophages [115,116]. The administration of an NMDAR antagonist decreases bone volume in mice, and the NMDAR antagonist inhibits osteoclast differentiation [117]. NMDAR affects bone homeostasis. Furthermore, plasmin attenuates LPS-induced osteoclastogenesis [66]. In contrast, tPA induces MMP-9 expression, and tPA-induced MMP-9 may affect osteoclastogenesis [118]. tPA may regulate osteoclastogenesis and inflammatory bone loss through plasmin-dependent and -independent mechanisms.

### 3.4. α2-Antiplasmin (α2AP)

α2AP, a member of the serine protease inhibitor (serpin) superfamily, is known to be a plasmin inhibitor. α2AP rapidly inactivates plasmin, forming a stable inactive complex [119]. PAP levels are elevated in patients with inflammatory diseases, including rheumatoid arthritis and diabetic nephropathy [120,121]. Additionally, PAP is associated with the secretion of IgG and IgM in human mononuclear cells [122]. α2AP inhibits plasmin by forming a stable 1:1 complex with plasmin through a two-step mechanism. Initially, the C-terminal end of α2AP, which contains six lysine residues, noncovalently binds to the lysine-binding sites (LBSs) present in the Kringle domains of plasmin. In the second step, the arginine residue at position 376 of α2AP in the reactive center loop forms a covalent bond with the active-site serine residue at position 741 of plasmin. This process results in the formation of the PAP complex, leading to a complete loss of plasmin activity [123]. On the other hand, the N-terminal sequence is crosslinked to fibrin by factor XIIIa. A protease, such as fibroblast activation protein (FAP) or antiplasmin-cleaving enzyme (APCE), causes the conversion of Met-α2AP to Asn-α2AP (12-amino-acid residue shorter form) [124,125]. α2AP is most closely related to the noninhibitory serpin pigment epithelium-derived factor (PEDF), and their structures are very similar [126,127]. α2AP can bind to the PEDF receptor (PEDFR) and adipose triglyceride lipase (ATGL)/calcium-independent phospholipase A2 (iPLA2) and regulate cytokine production (TGF-β, TNF-α, IL-1β), ECM production, as well as various cellular functions including differentiation and proliferation [128,129,130,131]. It is associated with various biological functions, including angiogenesis, inflammation responses, tissue remodeling, and the immune system [132,133,134,135]. 

The expression of α2AP has been detected in bone tissue in mice [136], and α2AP deficiency has been shown to enhance the bone formation rate in mice [137]. Additionally, α2AP deficiency attenuates ovariectomy (OVX)-induced trabecular bone loss in mice [138]. α2AP treatment induces the production of TNF-α and IL-1β through the ERK1/2 and p38 MAPK [134,138]. Moreover, α2AP negatively affects osteoblast differentiation and function by inhibiting the Wnt/β-catenin pathway [137]. α2AP also regulates the activation of tyrosine phosphatase SHP2 [135]. SHP2 regulates osteoclastogenesis by promoting preosteoclast fusion [139]. Furthermore, α2AP deficiency has been found to attenuate the progression of several inflammatory diseases such as lupus nephritis and diabetic nephropathy in mice [129,134]. The blockade of α2AP by a neutralizing antibody or by miRNA has been shown to attenuate fibrosis progression and vascular damage in mice [133,135,140]. The expression of α2AP is induced by pro-inflammatory mediators, such as CTGF, HMGB1, and IFN-γ [129,130,134]. CTGF promotes osteoclastogenesis by inducing and interacting with dendritic cell-specific transmembrane protein (DC-STAMP) [141]. Additionally, CTGF enhances RANKL-induced osteoclast differentiation through direct binding to RANK and OPG [142]. HMGB1 regulates osteoclastogenesis [143] and induces bone destruction through RAGE and TLR4 [144]. In contrast, IFN-γ inhibits osteoclastogenesis through downregulating NFATc1 and promotes osteoclast apoptosis [145]. Additionally, IFN-γ induces osteoblast differentiation [145]. CTGF, HMGB1, and IFN-γ-produced α2AP may affect osteoclastogenesis and bone destruction. Moreover, MMP-3 can degrade α2AP and inactivate α2AP [146]. The MMP3 inhibitor preserves osteoclast differentiation and survival in the presence of 17β-estradiol, demonstrating the necessity of MMP3 for 17β-estradiol-induced osteoclast apoptosis [147]. Furthermore, PEDF deficiency results in bone defects and frequent fracturing in mice [148], and PEDF enhances osteoblastic differentiation and osteoblastic mineralization, inducing sclerostin and other osteocyte gene expression through the ERK/GSK-3β/β-catenin signaling pathway [149,150]. PEDF also inhibits osteoclast function by regulating OPG expression, thereby contributing to the maintenance of bone homeostasis [151]. α2AP is structurally similar to PEDF and can bind to PEDFR, suggesting that it may exhibit similar effects as PEDF. On the other hand, α2AP deficiency promotes VEGF over-release in mice [132], and α2AP mediates the VEGF signal pathway through the ATGL/SHP2 axis [135]. Additionally, α2AP attenuates Wnt-3a-induced β-catenin expression and mediates Wnt/β-catenin signaling [137]. Moreover, α2AP is associated with apoptosis [152], suggesting that α2AP may affect osteocyte apoptosis through the regulation of multiple signaling pathways. Thus, α2AP may mediate bone remodeling and inflammatory bone loss through multiple mechanisms, including plasmin inhibition, pro-inflammatory cytokine production, and the regulation of several signaling pathways.

### 3.5. Plasminogen Activator Inhibitor-1 (PAI-1)

PAI-1, a member of the serpin superfamily, is recognized as an inhibitor of uPA and tPA, thereby regulating the activation of plasmin and the fibrinolytic system [153]. PAI-1 is a single-chain molecule with two interactive domains, including a surface-exposed reactive center loop (RCL). PAI-1 can interact with uPA and tPA, inhibiting plasmin-mediated fibrinolysis and proteolysis [154]. Additionally, PAI-1 has a flexible joint region that binds to non-proteinase ligands, such as Vn and members of the low-density lipoprotein receptor (LDLR) family [155]. The expression of PAI-1 is induced by various factors, including growth factors (TGF-β, IGF-1, and PDGF, and EGF), inflammatory cytokines (TNF-α and IL-1β), hormones (insulin, GC, and angiotensin II), and glucose [156,157,158,159,160]. 

PAI-1 has been detected in samples from patients with chronic inflammatory diseases, including rheumatoid arthritis, systemic lupus erythematosus, Crohn’s disease, and periodontitis (Figure 3). In PAI-1-deficient mice, the length and size of the femur are smaller than those of wild-type mice, resulting in an increase in total bone mineral density [161]. Regarding fracture healing, PAI-1-deficient mice developed larger and more mineralized fracture calluses than wild-type mice. Additionally, PAI-1 deficiency protects against streptozotocin (STZ)-induced bone loss in female mice [162]. Furthermore, osteoclast levels in the tibia are attenuated by PAI-1 deficiency in female mice injected with STZ [162]. PAI-1 deficiency also protects against trabecular bone loss in conditions of Es deficiency in mice [163]. Furthermore, PAI-1 deficiency attenuated GC-induced bone loss, presumably by inhibiting the apoptosis of osteoblasts [164]. Several studies have demonstrated that PAI-1 deficiency decreases aspects of the inflammatory response, such as neutrophil infiltration [165,166]. In contrast, PAI-1 stimulates the infiltration of inflammatory cells, including macrophages [167]. Furthermore, PAI-1 can interact with Vn and LDLR [155]. Vn plays an important role in tissue remodeling, cell migration, differentiation, and inflammation response and is associated with bone homeostasis. A Vn-derived peptide inhibits osteoclastogenesis by binding to c-Fms and inhibiting M-CSF signaling [168], and the administration of the Vn-derived peptide reversed ovariectomy-induced bone loss [169]. LDLR deficiency causes impaired osteoclastogenesis [170] and reduced bone mass through the c-fos/NFATc1 pathway in mice [171]. Additionally, LDLR family members mediate Wnt/β-catenin, TGF-β, bone morphogenetic proteins (BMPs), and PDGF signaling [172]. The interaction of PAI-1 with Vn or LDLR may affect osteoclastogenesis and bone homeostasis. Increased expression of PAI-1 by various cytokines and inflammatory mediators may promote an inflammatory response and induce osteoclastogenesis and bone destruction through the inhibition of plasmin functions and the promotion of various protein interactions involving PAI-1.

## 4. Conclusions and Therapeutic Perspective

Fibrinolytic factors, including Plg, plasmin, uPA, uPAR, tPA, α2AP, and PAI-1, are reportedly associated with bone homeostasis and inflammatory responses through immune cell activation, cytokine production, and the regulation of cell signaling. Several studies using various animal models, deficient mice, and patient samples with chronic inflammatory diseases suggest that fibrinolytic factors influence the progression of inflammatory bone destruction. However, the detailed mechanism by which fibrinolytic factors regulate the progression of inflammatory bone destruction has not been fully elucidated. Clarifying the detailed mechanism underlying the roles of fibrinolytic factors in bone metabolism and the inflammatory response and advancing related clinical research are expected to lead to a novel therapeutic approach for inflammatory bone diseases.

## Figures and Tables

**Figure 1 cells-13-00516-f001:**
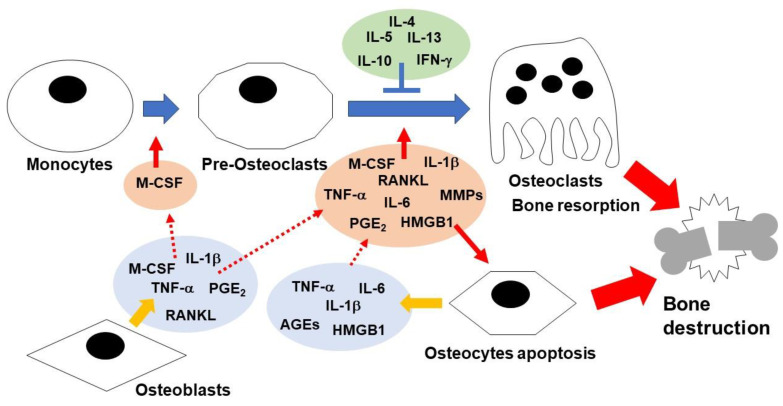
The role of bone cells in inflammatory bone destruction. Several pro-inflammatory cytokines, including TNF-α, IL-1β, and IL-6, promote osteoclastogenesis and osteocyte apoptosis. In contrast, IFN-γ, IL-4, IL-5, IL-10, and IL-13 inhibit osteoclastogenesis.

**Figure 2 cells-13-00516-f002:**
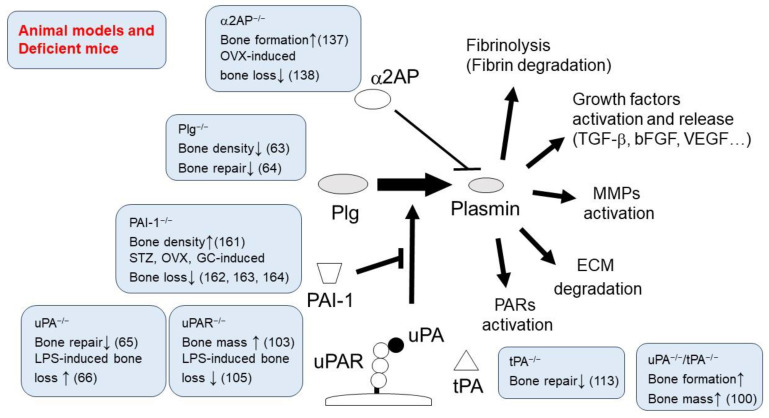
Key findings of the effects of fibrinolytic factors in bone remodeling and inflammatory bone loss. Fibrinolytic factors, including Plg, uPA, uPAR, tPA, α2AP, and PAI-1, have various functions other than fibrin degradation. Studies using fibrinolytic factor-deficient mice have shown that fibrinolytic factors are involved in bone metabolism and bone destruction.

**Figure 3 cells-13-00516-f003:**
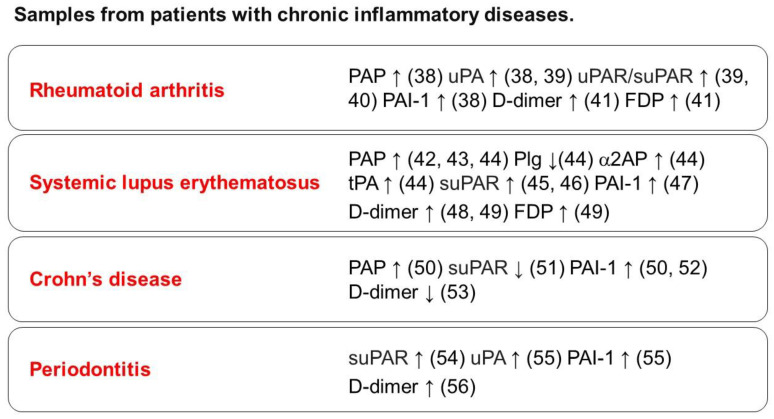
The levels of fibrinolytic factors in samples from patients with chronic inflammatory diseases.

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
