# Peer review of "The Roles of Fibrinolytic Factors in Bone Destruction Caused by Inflammation"

_cells, 2024, doi:10.3390/cells13060516_

Round 1

Reviewer 1 Report

Comments and Suggestions for Authors

Dear authors,

You have presented an excellent review.

Minor suggestions:

Please modify the first sentence of the Abstract as it mentions only rheumatoid arthritis and this is misleading to the readers as the focus is not only on RA. 

From the Conclusion, it is unclear what exactly this review's point is. The authors themselves say that the significance of these factors has already been established. The last sentence of the Conclusion, which should give the point of the whole article, actually stands as a helpless explanation of this review and is redundant. The last sentence of the Conclusion should be deleted or The Conclusion should be revised and present the meaning of this review and what it will contribute to the scientific community. 

The author cited 28 papers from his previous scientific work on the subject, which shows that the author is proficient in the subject. 

Author Response

Thank you for your important comments.

As you pointed out, I corrected the first sentence of the Abstract as follows:

Chronic inflammatory diseases, such as rheumatoid arthritis, spondylarthritis, systemic lupus erythematosus, Crohn’s disease, periodontitis, and carcinoma metastasis frequently result in bone destruction. 

In addition, as you pointed out, I rewrote the conclusion as follows.

Fibrinolytic factors, including Plg, plasmin, uPA, uPAR, tPA, α2AP, and PAI-1, are reportedly associated with bone homeostasis and inflammatory responses through immune cell activation, cytokine production, and the regulation of cell signaling. Several studies using various animal models, deficient mice, and patient samples with chronic inflammatory diseases suggest that fibrinolytic factors influence the progression of inflammatory bone destruction. However, the detailed mechanism by which fibrinolytic factors regulate the progression of inflammatory bone destruction has not been fully elucidated. Clarifying the detailed mechanism underlying the roles of fibrinolytic factors in bone metabolism and inflammatory response, and advancing related clinical research, is expected to lead to a novel therapeutic approach for inflammatory bone diseases.

Reviewer 2 Report

Comments and Suggestions for Authors

Dear Authors,

You raised interesting topic, important in the field.

-However your manuscript is difficult to follow. I strongly suggest the reedition of the text. In my opinion in many parts of the text sentences are too long. For example in lines 115-126 there is one sentence which is really difficult to digest. Similar edition is repeated in many parts of the manuscript

-Figures should be more clearly explained

Author Response

Thank you for your important comments.

As you pointed out, I restructured the sentences and shortened the sentences that were too long.

In addition, as you pointed out, I added the comment in Figures as follows:

Figure 1. The role of bone cells in inflammatory bone destruction.  Several pro-inflammatory cytokines, including TNF-α, IL-1β, and IL-6 promote osteoclastogenesis and osteocytes apoptosis.  In contrast, IL-4, IL-5, IL-10, IL-13, and interferon-γ (IFN-γ) inhibit osteoclastogenesis.

Figure 2. Key findings of the effects of fibrinolytic factors in bone remodeling and inflammatory bone loss.  Fibrinolytic factors, including Plg, uPA, uPAR, tPA, α2AP, and PAI-1 have various functions other than fibrin degradation.  Studies using fibrinolytic factors-deficient mice have shown that fibrinolytic factors are involved in bone metabolism and bone destruction.

Reviewer 3 Report

Comments and Suggestions for Authors

The Review by Yosuke Kanno is focused on the role of fibrinolytic factors in inflammatory bone destruction and it is potentially very  interesting, but I understand, as stated by the author in the last paragraph ("Conclusion and therapeutic perspective"), that the mechanisms and the activities elicited by the fibrinolytic factors were mainly analyzed in animal models. How many of the reported results were obtained in humans? In in vitro human cells and in in vivo, in human serum or plasma? 

At the moment, the review seems to be a mixture of murine and human data and informations and this could create in readers misunderstanding or an excessive extrapolation and generalization of the results. Instead, I think it's very crucial understand what happens in humans, mainly in order  to think farmacological or therapeutic approaches. So, in my opinion, the review will acquire more significance and value as author will introduce in each section (3.1, 3.2, 3.3, etc...) a comparison of murine versus human inflammatory  bone destruction and/or a Table to list and explain the similar/different pathways/mechanisms activated/inhibited in humans versus muose/rat systems. Indeed, it's very difficult to think novel therapeutic approaches for human inflammatory bone diseases without a clear vision of the state of the art in human.

Another point is the very high number of autocitations: please, reduce them.

Author Response

Thank you for your important comments.

As you pointed out, I added the comment for several clinical research follows:

The levels of fibrinolysis markers such as D-dimer and fibrin degradation products (FDP), as well as fibrinolytic factors in samples from patients with chronic inflammatory diseases like rheumatoid arthritis (39-42), systemic lupus erythematosus (43-50), Crohn’s disease (51-54), and periodontitis (55-57), are higher than those in healthy controls (Fig. 3). These factors are associated with the pathology of chronic inflammatory diseases and may play a critical role in the context of inflammatory bone destruction.

Additionally, as you suggested, I presented the animal data and patient sample data separately in Figures 2 and 3 to prevent misunderstandings among readers.  Moreover, I added whether some studies used animal samples or patient samples.

Furthermore, as you pointed out, I reduced autocitations. 

Round 2

Reviewer 2 Report

Comments and Suggestions for Authors

Thank you for your corrections. I have no further concerns

Author Response

Thank you.

Reviewer 3 Report

Comments and Suggestions for Authors

The work is accepted.

Author Response

Thank you.